# Impact of Variation in Commissural Angle between Fused Leaflets in the Functionally Bicuspid Aortic Valve on Hemodynamics and Tissue Biomechanics

**DOI:** 10.3390/bioengineering10101219

**Published:** 2023-10-18

**Authors:** Elias Sundström, Justin T. Tretter

**Affiliations:** 1Department of Engineering Mechanics, Flow Research Center, KTH Royal Institute of Technology, Teknikringen 8, 100 44 Stockholm, Sweden; 2Congenital Valve Procedural Planning Center, Department of Pediatric Cardiology and Division of Pediatric Cardiac Surgery, Cleveland Clinic Children’s, and The Heart, Vascular, and Thoracic Institute, Cleveland Clinic, Cleveland, OH 44195, USA; trettej3@ccf.org

**Keywords:** bicuspid aortic valve, commissural angle, fluid–structure interaction, magnetic resonance

## Abstract

In subjects with functionally bicuspid aortic valves (BAVs) with fusion between the coronary leaflets, there is a natural variation of the commissural angle. What is not fully understood is how this variation influences the hemodynamics and tissue biomechanics. These variables may influence valvar durability and function, both in the native valve and following repair, and influence ongoing aortic dilation. A 3D aortic valvar model was reconstructed from a patient with a normal trileaflet aortic valve using cardiac magnetic resonance (CMR) imaging. Fluid–structure interaction (FSI) simulations were used to compare the effects of the varying commissural angles between the non-coronary with its adjacent coronary leaflet. The results showed that the BAV with very asymmetric commissures ($120^{\circ }$ degree commissural angle) reduces the aortic opening area during peak systole and with a jet that impacts on the right posterior wall proximally of the ascending aorta, giving rise to elevated wall shear stress. This manifests in a shear layer with a retrograde flow and strong swirling towards the fused leaflet side. In contrast, a more symmetrical commissural angle ($180^{\circ }$ degree commissural angle) reduces the jet impact on the posterior wall and leads to a linear decrease in stress and strain levels in the non-fused non-coronary leaflet. These findings highlight the importance of considering the commissural angle in the progression of aortic valvar stenosis, the regional distribution of stresses and strain levels experienced by the leaflets which may predispose to valvar deterioration, and progression in thoracic aortic dilation in patients with functionally bicuspid aortic valves. Understanding the hemodynamics and biomechanics of the functionally bicuspid aortic valve and its variation in structure may provide insight into predicting the risk of aortic valve dysfunction and thoracic aortic dilation, which could inform clinical decision making and potentially lead to improved aortic valvar surgical outcomes.

## 1. Introduction

A functionally bileaflet or bicuspid aortic valve (BAV) with a trisinuate aortic root is a common congenital heart condition that presents challenges in diagnosis and treatment due to its variable progression of valvar dysfunction and thoracic aortic dilation [1,2]. The severity of aortic valvar stenosis is known to be related to the size of the valvar opening. Proper leaflet coaptation is influenced by the relative dimensions of the leaflets in relation to the dimensions of the planes throughout the aortic root, from the virtual basal ring to the sinutubular junction. However, the role of variation in the position of the two commissures present in this most common form of a bicuspid aortic valve, or commissural angle, remains unclear in its impact on valvar function [3,4,5].

Recently, a study using cardiac magnetic resonance (CMR) imaging and fluid–structure interaction (FSI) simulations investigated the impact of the height of the interleaflet triangle or commissural height on the degree of aortic valvar stenosis, hemodynamics, and tissue biomechanics. The study found that a larger zone of fusion in the functionally BAV with an inversely reduced interleaflet triangle height resulted in a linear rise in wall shear stress, peak velocity, pressure gradient, and strain levels, forming more asymmetric vortex systems and the recirculation of flow toward the side of leaflet fusion within the trisinuate aortic root. The study’s findings highlight the importance of considering the interleaflet triangle height as a crucial factor in the development of thoracic aortic dilation in patients with BAV, along with potential considerations related to durable aortic valvar repair [6].

Flow MRI has also been used to investigate how the degree of asymmetry of the BAV, with commissural angles between 120 and 180${}^{\circ }$, affects the outflow jet as compared to the normal trileaflet aortic valve (TAV) [7]. The blood flow was considered during peak systole and its implications for aortopathy (aortic disease). The results showed that asymmetric BAVs had eccentric outflow jets that affected specific regions of the aortic wall based on the position of the smaller leaflet. In contrast, symmetric BAVs had more centered outflow jets that did not impact the aortic wall. The symmetry of the BAV and the position of the smaller leaflet were key factors influencing the outflow jet characteristics. However, there was no quantification of stresses and strain levels in the leaflet material that may predispose to aortic valvar dysfunction.

This current study aims to build upon previous research [6,8,9,10] and investigate the resulting hemodynamic and tissue biomechanical impact related to variation in the commissural angle found in the functionally bileaflet aortic valves with a trisinuate aortic root (Figure 1). Like the variation in the degree of fusion between the two fused leaflets in this common form of a bicuspid aortic valve, the variation described in the commissural angle is believed to affect the systolic valvar opening area and the interplay between hemodynamics and biomechanical responses in the thoracic aorta. Therefore, this study will complement the previous assessments of the effects of normal variation in the rotation position of the aortic root relative to the base of the left ventricle and the degree of leaflet fusion [10], as well as the variation of the interleaflet triangle height [6].

## 2. Method

### 2.1. CMR

Cardiac magnetic resonance (CMR) scans were performed at Cincinnati Children’s Hospital Medical Center (CCHMC) on an adolescent subject who had normal cardiovascular anatomy and function, see [9,10]. All demographic information, such as age, weight, gender, and diagnosis, was de-identified. The study was therefore deemed exempt from the ethical review and approval by the Cincinnati Children’s Institutional Review Board.

The CMR acquisition protocol used a 1.5 T CMR machine (Ingenia, Philips Healthcare, Best, the Netherlands) equipped with a phased-array coil. The protocol incorporated various imaging sequences, including the axial aortic root cine stack, phase-contrast velocity sequence, and non-contrast 3D mDixon angiogram. The short-axis aortic root cine stack was obtained using a steady-state free precession pulse sequence. This sequence employed a repetition time of 7.8 ms, an echo time of 4.7 ms, a flip angle of 15 degrees, and sequential 2 mm slices with no interslice gap. To capture the phase-contrast velocity information, a gradient-echo sequence was employed with a repetition time of 4.3 ms, an echo time of 2.7 ms, a flip angle of 12 degrees, and a slice thickness of 6 mm. The encoding velocity used was 1.5 m/s. The scanning protocol achieved a mean time resolution of 30–40 ms, resulting in 30 phases per cardiac cycle. In the coronal plane, a non-contrast 3D mDixon angiogram was acquired, using a repetition time of 5.3 ms, a flip angle of 15 degrees, and a spatial resolution of 1 mm. The details of the aortic root were assessed by analyzing the information obtained from the aortic root cine steady-state free precession sequence, see [9,10]. In addition, the protocol for the long-axis sagittal stack 4D Flow MRI was acquired using a velocity encoding of 2 m/s, a repetition time of 3.5 ms, an echo time of 1.9 ms, and a flip angle of 8°, see reference [8].

### 2.2. Aortic Reconstruction

Geometry reconstruction of the subject’s aortic valve and thoracic aortic anatomy was performed using the segmentation software 3D Slicer 5.2.2 [11]. The outlines of the aortic fluid domain were defined by specifying a threshold intensity on the 3D mDixon angiogram CMR dataset. The resulting control volume encompassed the aortic sinuses, thoracic aorta, and the head and neck vessels, including the right brachiocephalic, left common carotid, and left subclavian arteries. High-curvature features were captured using fine, high-quality tessellation applied to the control surfaces, which was followed by a smoothing protocol to reduce irregularities.

A short distance upstream of the aortic root, the inlet surface was extruded, and a uniform blood flow velocity was specified using the flow rate shown in Figure 2 [12]. This allowed for the development of a velocity profile with a boundary layer inferiorly to the aortic valve.

Leaflet attachment lines and commissures observed in the CMR images were used to reconstruct anatomically accurate semilunar contact between the leaflets and the walls of the aortic sinuses. The spatial resolution of the CMR images facilitated the identification of the leaflets’ semilunar features. The reconstructed leaflet surface was then inflated to a thickness of 0.7 mm, representing the lower end of a multi-ethnic population dataset [13], thus forming the solid domain of the leaflet tissue. However, due to the limited spatial resolution of the CMR, some high-curvature features of the aortic leaflets were not captured. To aid the reconstruction process, the remaining leaflet attachment lines, which were not visible in the CMR images, were specified using ratios derived from an averaged homograft dataset from a population with normal aortic valves [14].

The most common type of bicuspid aortic valve is characterized by fused leaflets, resulting in a functional bileaflet valve within a trisinuate aortic root. There is a variable degree of a raphe at the zone of fusion. This form is present in 90–95% of individuals with a bicuspid aortic valve [4,5]. The functionally bileaflet phenotypes include right–left leaflet fusion, right–non-leaflet fusion, and rarely left–non-leaflet fusion. Figure 3 illustrates long-axis and short-axis views of the complex 3D structure of the aortic root and its leaflets, similar to 3D CT and MRI. In all phenotypes shown, three well-defined aortic sinuses are evident, with variability in the degree of fusion between the coronary leaflets.

The functionally bileaflet aortic valve is further characterized by variation in the angle between its two commissures. The short-axis view in Figure 3 displays different angles of the commissures. The functionally bileaflet aortic valve with commissural symmetry is defined by a commissural angle of 160–180 degrees; that with asymmetrical commissures exhibits an angle of 140–159 degrees; and that with very asymmetrical commissures exhibits an angle of 120–139 degrees.

### 2.3. Computational Models

Similar to previous studies [6,8,9,10], we took into account the conservation of mass and momentum laws to simulate the transport of blood flow through the aorta. We described the non-Newtonian behavior of blood using a mass transport equation for the volume fraction between red blood cells (RBCs) and blood plasma. The density of the blood was modeled as a linear combination of the densities of RBCs and blood plasma. The diffusion rate of RBCs was governed by a parameter that depends on the viscosity and mass diffusivity. To incorporate rheology effects, we used a modified power law approach based on the Ostwald–de Waele rheology model. This modification aimed to adjust the viscosity to match the viscosity of blood plasma in scenarios with zero shear rate and zero RBC volume fraction. Empirical correlations were used to determine the model parameters [15].

The flow variation at the inlet boundary was measured with CMR phase contrast data during the cardiac cycle, see Figure 2. For the head and neck vessels, we employed a Windkessel three-element circuit function to connect the flow rate and pressure. We used a numerical scheme to obtain the pressure level at each time step, considering the physiological pressure ratios. The proximal resistance (Rp), distal resistance (Rd), and compliance (C) in the Windkessel model were tuned for physiological pressures of 120 mmHg at peak systole and 80 mmHg at the end of diastole. The governing equation and constitutive relations were solved using the finite volume method, with a Rhie and Chow-type velocity coupling and a semi-implicit method for pressure-linked equations [16,17,18,19].

The solid tissue was modeled as a nearly incompressible hyperelastic material using the Ogden model [20,21]. The elastic properties were determined through an inverse optimization-based process to match the displacement of the leaflet edge with CMR data. The model was constrained by fixed support at the interface between the leaflets and the aortic root, whereas all other surfaces were free to move. The exchange of information between the fluid and solid domains was governed by a fluid–solid contact interface. The stiffness matrix was updated using Newton iteration methods to handle nonlinear material specifications and large deformations. Four wedge layers discretized the leaflets, whereas polyhedral cell and prism layers discretized the fluid domain. Grid convergence studies were performed in previous studies to ensure accurate results [10]. The FSI model used in previous studies was validated by comparing the velocity distribution between CMR data and numerical results for the baseline case [9,10]. The computed flow field showed good agreement with the literature [12,22,23,24], with a difference of about 15%. The simulations were performed in parallel on 112 cores (Intel Xeon E5-2680), each case taking approximately six hours for one cardiac cycle.

## 3. Result

Figure 4 presents a comprehensive depiction of blood flow through the aorta and the aortic valve during various stages of the cardiac cycle for the normal trileaflet aortic valve (TAV) and the functionally bileaflet aortic valve (BAV). During early systole (t/T=0.04), the aortic valve initiates its opening, leading to the ejection of blood from the left ventricle. In the narrower section of the valve, local flow velocity begins to accelerate, and a jet of flow forms through the valve in both TAV and BAV cases with approximately stagnant flow in the ascending aorta. In the next time instant (t/T=0.07), the aortic valve is now half open, where the BAV cases show severe stenosis on the side with the fully fused left and right leaflets, limiting the displacement as compared to the normal TAV case. The streamlines in Figure 5 in both TAV and BAV cases during early systole are directed towards the convex side of the ascending aorta, which is consistent with the curvature of the ascending aorta.

Around the time of peak systole (between t/T values of 0.1 and 0.2), the velocities in the ascending aorta increase, corresponding to the pulse wave velocity assessment and the time delay for the pulse wave to propagate through the aorta, see Figure 2. In the TAV case, the velocity vectors remain relatively aligned with the aorta. However, in the BAV cases, there is a higher peak velocity that impacts more towards the convex tissue wall near the sinotubular junction and proximally in the ascending aorta compared to the TAV case. The velocity field depicts a strong shear layer with a recirculating flow towards the concave side of the ascending aorta, which is in good agreement with a previous blood speckle imaging study [25]. The shear layer intensifies, and during post-peak systole (between t/T values of 0.2 and 0.3), it occupies a significant portion of the region, extending from the aortic root to the proximal aortic arch and its right brachiocephalic and the left common carotid arteries. The streamlines in the short-axis cut of the ascending aorta at peak systole for the TAV case show two counter-rotating vortices, see Figure 5. These are Dean-like vortices that develop due to the curvature of the ascending aorta. In the BAV cases, there are also counter-rotating vortices but with an asymmetry compared to the TAV case [26]. Towards post-peak systole, the TAV case depicts three coherent vortices that coincide with the apexes of the three commissures at the level of the sinutubular junction. In the BAV case, towards post-peak systole, the streamlines depict a swirling flow with local incoherent vortices located between the shear-layer and circulation zone.

As the systolic phase concludes (between t/T values of 0.2 and 0.3), the valve closes, leading to a decrease in blood flow concurrent with a reduction in aortic pressure during diastole, see last time instant in Figure 4 and Figure 5. As time progresses towards diastole, the flow will settle down due to diffusion, promoting the filling of the left ventricle with fresh blood in preparation for another systole. The streamlines on the short-axis cut for the BAV case exhibit residual flow with a small swirl, indicating a longer diffusion time compared to the TAV case.

Figure 6 quantifies the streamwise and cross-flow velocity profiles on the short-axis cut plane, c.f. Figure 5 for the location of the profile. For the TAV case, the streamwise velocity (Figure 6a) indicates a top hat distribution with a slight slope towards the convex side of the ascending aorta. There is a fair degree of agreement with the 4D Flow MRI data, where the difference in the peak velocities is within 15%. The BAV cases show higher velocities towards the convex side of the ascending aorta. There is a general trend of an increasing velocity gradient with a reduced commissural angle. On the opposite side, there is a retrograde flow, i.e., towards the side with the RL leaflet fusion. It is observed that the shear layer shifts towards the fusion side, and the magnitude of the flow reversal reduces with an increasing commissural angle. The cross-flow component (Figure 6b) for the TAV case indicates little to no swirl, which is in fair agreement with the 4D flow MRI. However, all BAV cases indicate a notable cross flow that correlates with a strong swirling component, c.f. Figure 5. There is a general trend that a reduced commissural angle increases the tangential velocity gradient on the convex side of the ascending aorta. It is also observed that the magnitude or the cross flow reduces with an increased commissural angle.

Figure 7a shows the aortic opening area for the TAV case compared with the BAV case with fully fused coronary leaflets for three different commissural angles ($120^{\circ }$, $150^{\circ }$, $180^{\circ }$). Both the TAV and BAV cases open around t/T=0.05. The opening is nearly linear until the peak opening, around t/T=0.1. The TAV case shows a normal aortic opening area around 4 cm${}^{2}$, whereas the BAV cases are stenoic with an aortic opening area below 2 cm${}^{2}$. There are only trivial to no oscillation wiggles, indicating critical damping. All cases show a non-symmetric top-hat-like distribution with more rapid opening than closing. In addition, the TAV case closes earlier than the BAV cases, which correlates with a higher LV pressure to produce the same flow rate.

Figure 7b,c depict the radial displacement of the non-coronary and right coronary leaflets, i.e., the free edge’s mid-point. The TAV case shows an asymmetric opening where the right coronary leaflet exhibits a larger radial displacement than the non-coronary leaflet, which is probably due to the normal minimal asymmetries present in the geometry of the valvar leaflets. For the BAV cases, there is a linear trend with increasing radial displacement of the unfused non-coronary leaflet as a function of the commissural angle with a decreasing commissural angle. Specifically, the BAV $120^{\circ }$ deg case has a radial displacement of the non-coronary leaflet that is twice as large compared to that of the BAV $180^{\circ }$ case. It is also observed that the BAV $180^{\circ }$ case opens up earlier than the BAV $120^{\circ }$ case. When considering the radial displacement of the fused right leaflet there is also a similar linear trend but reversed, where the radial displacement increases with an increased commissural angle. However, this trend is not as clear compared to the radial displacement of the unfused non-coronary leaflet.

Figure 8 shows the von Mises stress distribution for the TAV and BAV cases during different stages of the cardiac cycle. At the beginning of the valve opening t/T=0.04, all cases shows stress levels below 0.04 MPa. Between t/T=0.07 and t/T=0.021, the stress level gradually increases, where elevated stress levels are present at the interface of the leaflets with the aortic root and near the leaflet free edge. Upon the valve closure, the entire valve is pushed downwards due to the adverse pressure gradient, and the stress level gradually increases with concentrated levels at the interface of the leaflets with the aortic root but also in the mid-portion of the leaflet surface.

Figure 9 shows the Frobenius norm of the strain tensor for the same cases and time instant as in Figure 8. At the beginning of the opening t/T=0.04, all cases shows low strain levels, indicating that the valve is close to its neutral stress-free configuration. Between time instants t/T=0.07 and t/T=0.021, the strain level increases, with the TAV case showing larger strain due its larger displacement and aortic opening area, c.f. Figure 7. Similar to the quantification of the radial displacement of the unfused non-coronary leaflet, i.e., Figure 7b, there is a reduced strain as a function of the commissural angle. However, the strain on the fused side of the BAV does not show a linear variation as a function of the commissural angle. Instead, the fused RL leaflet for the BAV $180^{\circ }$ depicts a growing strain concentrated along the fold that starts close to the location of the free edge of the zone of fusion and its raphe, and extending across the midline of both fused leaflets towards their respective nadirs. As time evolves to time instant t/T=0.52, the valve closes, and the strain level gradually increases in the mid-portion of the leaflets due to the increasing adverse pressured gradient during diastole.

The stress and strain levels shown previously are now further quantified along the vertical intersection of the non-coronary leaflet, see Figure 10 (see black line annotations in Figure 8 for the location of the vertical intersection). At peak systole (t/T=0.1), the stress and strain levels are similarly distributed along the non-coronary leaflet for the TAV and the BAV $120^{\circ }$ deg cases, see Figure 10a,b. The stress shows a minimum at a location around 20–30% of the normalized effective leaflet height *s* and a maximum at the intersection of the leaflet with the aortic root (s=0). As the commissural angle increases, in the BAV $150^{\circ }$ deg and $180^{\circ }$ deg cases, both stress and strain levels reduce.

## 4. Discussion

This FSI analysis examines the influence of variation in the commissural angle in the most common BAV, the functionally bileaflet valve with trisinuate aortic root with a fusion between the left and right coronary leaflets. The study reveals the significance of this morphological feature in determining the resulting hemodynamics and tissue biomechanics. The decreasing aortic valve opening area seen in the spectrum from TAV to BAV with partial fusion, and to BAV with full fusion leads to increasing elevation in the blood flow velocity and pressure gradient, with an increasing wall shear stress seen along the convex surface of the proximal ascending aorta. It was found that the commissural angle significantly influences the aortic valvar outflow jet, where a highly asymmetric BAV leads to increased velocity gradient and wall shear stress on the posterior convex side of the ascending aorta. It is also evident that a BAV with very asymmetric commissures introduces a strong swirling flow during peak systole. These results suggest that the behavior of the aortic valvar outflow jet is influenced by both the symmetry and position of the smaller leaflet in BAVs, which is in good agreement with a previous 4D flow MRI study [7]. Conversely, BAVs with symmetrically positioned commissures showed mildly eccentric aortic valvar outflow jets at peak systole that did not impinge on the aortic wall. In BAVs with right and left coronary leaflet fusion, increasing the commissural angle will correspond to an increase in the size of the non-fused non-coronary leaflet. This correlates with decreasing stress and strain levels on the non-coronary leaflet (recall Figure 10). This result is not evident from previous studies relying on only 4D Flow MRI. With increasing the commissural angle, and hence decreasing the size of the fused coronary leaflets, there is not a clear trend in the stress and strain levels placed on the fused coronary leaflets.

The augmentation in the peak systolic velocity is intuitively linked to the valvar opening area, which corresponds to the degree of left and right coronary leaflet fusion. The alteration in stress, strain, and flow patterns is also influenced by the commissural angle. In this analysis, the commissural angle primarily affects the unfused non-coronary leaflet. Increasing the commissural angle leads to an increased surface area of the non-coronary leaflet but also reduced curvature at the intersection with the aortic root. This directs the aortic valvar opening area towards the center reducing the wall shear stress on the proximal ascending aorta.

Compared to the less common true BAV that is bileaflet with a bisinuate root, the functionally BAV and trisinuate aortic root assessed in this study is more commonly associated with aortic valvar stenosis and aortic dilation [27]. In the latter type, two functional commissures are present, with two normal underlying interleaflet triangles, both extending to the level of the sinutubular junction (Figure 3). In contrast, the interleaflet triangle inferior to the zone of fusion is hypoplastic, with no functional commissure, with its apex falling short of the sinutubular junction. These data support the idea that, in the functionally bicuspid aortic valve, variation in the commissural angle may impact the regional distribution of leaflet stresses and strains, which may predispose to ongoing leaflet thickening, sclerosis, and eventual calcification. This may in turn impact the progression of aortic valvar stenosis. Moreover, the study shows how changes in the commissural angle directly relate to disruptions in hemodynamics and tissue biomechanics within the aortic root and subsequent thoracic aorta. This understanding may further illuminate risk factors for the progression of aortic dilation.

These findings carry implications for evaluating congenitally malformed valves and guiding surgical repairs. To best achieve durable repairs, it was demonstrated that the commissural angle, whether symmetric versus very asymmetric, may provide guidance in maintaining a functionally bileaflet valve versus reconfiguration to a trileaflet valve, respectively [22,28]. While this general approach has been scientifically validated [22], the hemodynamic mechanisms underlying these successes have not been elucidated. Specifically, our results demonstrate that the non-fused leaflet in the functionally BAV has decreased stress and strain when larger, in the setting of symmetrical commissures. This may support why an approach leaving this variation functionally bileaflet leads to a more durable repair. Similarly, the non-fused leaflet experiences varying stress and strain when smaller in the setting of very asymmetric commissures in a functionally BAV. This suggests why converting this type to a trileaflet valve may be the preferred surgical approach. Prior investigations into computational fluid dynamics in BAV have predominantly focused on altered hemodynamics arising from the leaflet orientation and valve opening area [23]. Although these impact the resultant hemodynamics and tissue biomechanics, they merely scratch the surface of the intricate three-dimensional variation of the aortic root and its valve. This study adds to existing knowledge by demonstrating the substantial effects of the variation seen in the commissural angle of the most common phenotype of the functionally BAV.

The study findings suggest that clinical assessment of the affected commissural angles could enhance the understanding of the projected progression of aortic valvar stenosis and thoracic aortic dilation. In addition, an improved understanding of the impact of the commissural angle on the resulting hemodynamics and tissue biomechanics may help better fine-tune surgical repair approaches in the functional BAV. These findings require clinical validation to understand the progression of aortic valvar stenosis and aortic root and ascending aortic dilation, along with the impact of valvar repair durability.

## 5. Limitations

The study used data from a normal TAV subject and subsequently simulated variations in the commissural angle in a BAV with fusion between the right and the left leaflets. This approach offers a notable advantage for a small-scale study by isolating the studied variable and additional variables introduced by dissimilarities observed in the geometry and dimensions of the thoracic aorta in both normal and congenitally malformed aortic roots.

This investigation centered on the functionally BAV with fusion involving the coronary leaflets. Future research is required to investigate other phenotypes of both functionally BAVs.

In the simulation, the inlet and outlet stations of the thoracic aorta were considered fixed supports, and the contact between the leaflets and the thoracic aorta was modeled as solid. While the motion of the heart between systole and diastole induces cyclical displacement at these aortic stations, the diastolic displacement of the aortic root and velocity were documented at around 1 cm and 10 cm/s, respectively, using echocardiography and tissue Doppler velocimetry [29]. Similar assessments of aortic root motion have been conducted using CMR [30]. Given that the thoracic aorta’s length and the rapidity of jet development during valvar opening differ by an order of magnitude, aortic root motion might not significantly impact stress levels in the ascending aorta. However, challenges persist due to the low signal-to-noise ratio and image quality of temporally varying 3D echocardiography and CMR, which hamper the accurate quantification of the aortic root. This aspect is currently the subject of intense research and will be addressed in subsequent studies.

Acknowledging certain assumptions, the FSI analysis in this study encompassed (a) a uniform aortic leaflet thickness, (b) uniform material properties of the aorta, (c) fixed spatial support of the aorta, and (d) the absence of aortic root pull and twist during the cardiac cycle. Naturally, these parameters exhibit variations based on age and gender. Yet, refining these parameters through CMR techniques remains challenging due to the inherent limitations of the low signal-to-noise ratio and image quality.

## 6. Conclusions

Variation in the commissural angle of the functionally BAV impacts the stress and strain of both the non-fused and fused leaflets. This variation also influences the hemodynamics and tissue biomechanics experienced in the subsequent thoracic aorta. This may influence the progression of aortic valvar stenosis and aortic root and ascending aortic dilation, along with the durability of valvar repair strategies. Clinical studies are warranted to validate these findings and determine the utility of assessing and surgically manipulating this variation in commissural angle in the functionally BAV.

## Figures and Tables

**Figure 1 bioengineering-10-01219-f001:**
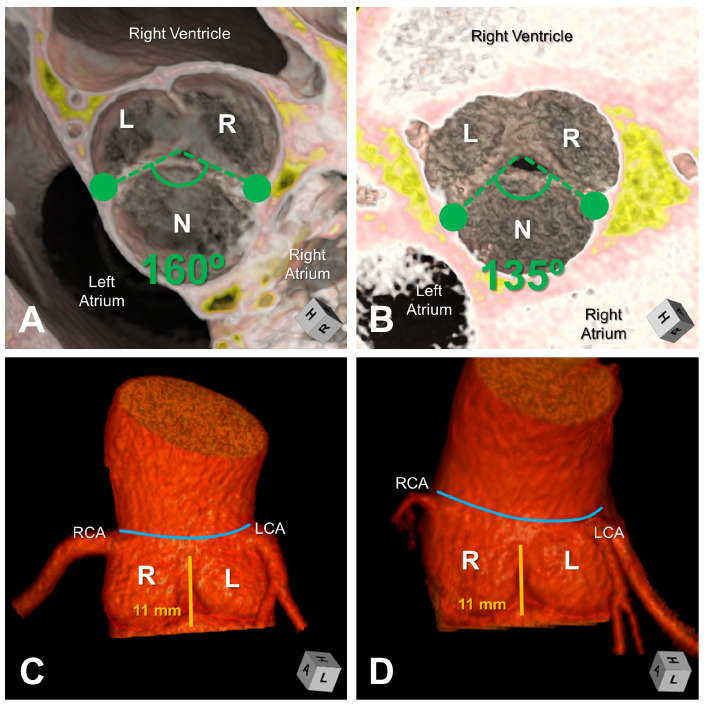
Computed tomographic 3D reconstructions of two patients with functionally bileaflet aortic valves with fusion between the coronary leaflets and trisinuate aortic roots are demonstrated (Patient 1 = Panels (**A**,**C**); Patient 2 = Panels (**B**,**D**)). Panels (**A**,**B**) demonstrate a short axis view of the aortic valve with the angle between the two commissures measured to be symmetrical and very asymmetrical, respectively. Prominent raphes are visualized at the zone of fusion between the coronary leaflets for both valves. Panels (**C**,**D**) demonstrate the blood-filled trisinuate aortic roots, both with similar commissural heights (11 mm) of the hypoplastic interleaflet triangle under the zone of fusion between the coronary leaflets. The sinutubular junction is marked with a blue line. L, left coronary sinus; LCA, left coronary artery; N, non-coronary sinus; R, right coronary artery; RCA, right coronary artery.

**Figure 2 bioengineering-10-01219-f002:**
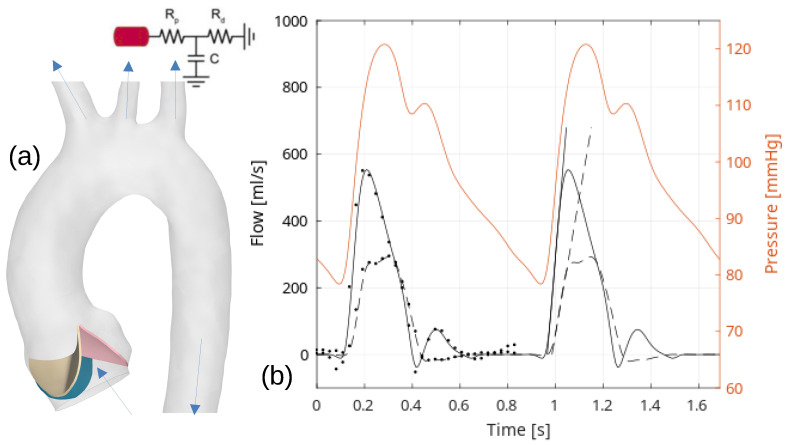
(**a**) Reconstructed geometry of the thoracic aorta is shown to the left, complete with specified boundary conditions (transparent grey color), and with the aortic valvar leaflets: left coronary (L), non-coronary (N), and right coronary (R). (**b**) Graphs in this section depict the flow rate during the cardiac cycle, both at the inlet and outlet (in the descending aorta). The pressure boundary conditions at the head and neck vessels across the cardiac cycle are established using the three-parameter Windkessel model. The solid surfaces of the leaflets that intersect with the aortic root are fixed, whereas all other solid surfaces are free.

**Figure 3 bioengineering-10-01219-f003:**
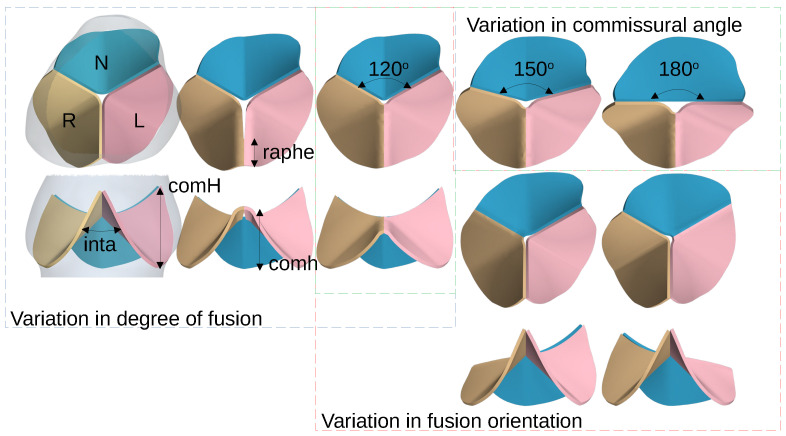
Within the blue hashed box, we have short-axis and long-axis reconstructed geometry of the normal trileaflet aortic valve, and with two variations in the extent of the zone of fusion, one with 2/3rd commissural height (comH) and partial fusion between the coronary leaflets, and the other with 1/3rd commissural height and complete fusion between the coronary leaflets. With decreasing commissural height, there is increased fusion and angle of the apex of the interleaflet triangle (inta). Within the green hashed box, there is a variation of the commissural angle, i.e., $120^{\circ }$, $150^{\circ }$, and $180^{\circ }$ deg. This increases the surface area of the unfused non-coronary leaflet. Within the red hashed box, there is a variation in the fusion orientation with the RN and rare LN leaflet fusion phenotypes.

**Figure 4 bioengineering-10-01219-f004:**
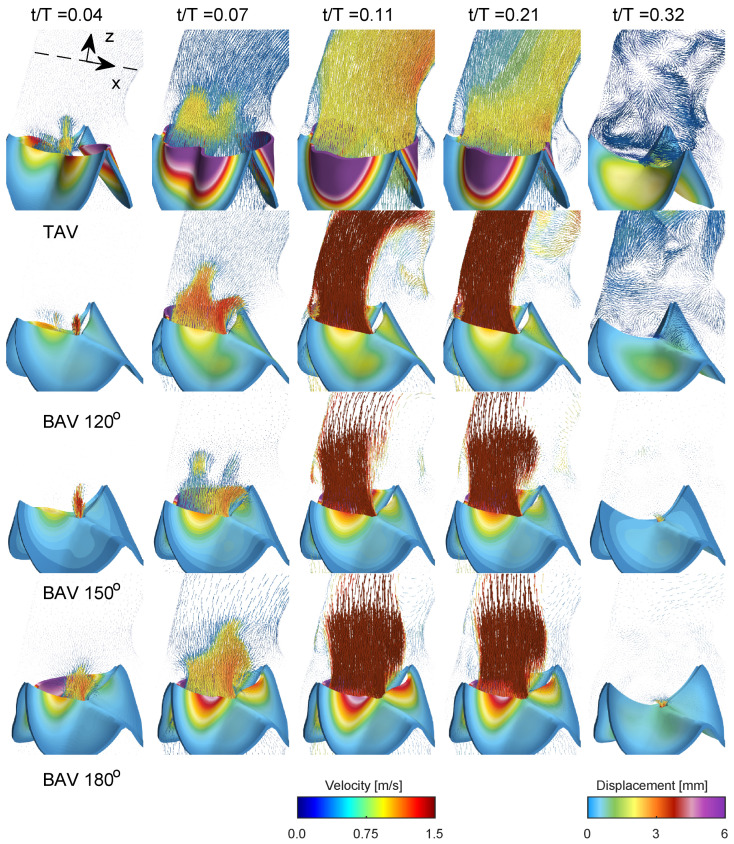
Blood flow visualization using velocity vectors at different time instants during the cardiac cycle for the TAV case and the BAV cases with fully fused coronary leaflets. The velocity vectors are presented on a vertical plane, coronally oriented.

**Figure 5 bioengineering-10-01219-f005:**
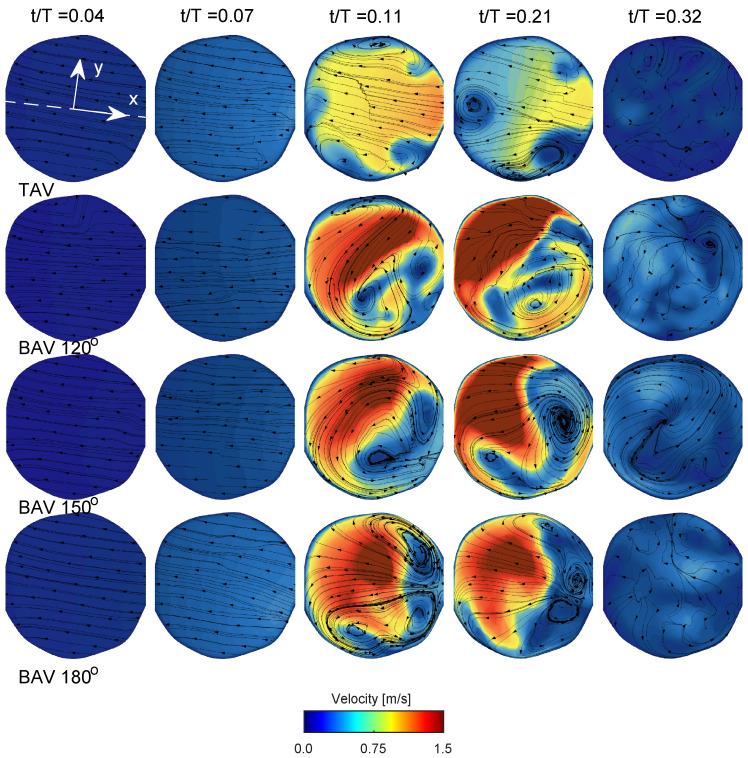
Velocity and streamline distribution shown on the short-axis cut plane. The plane is located 1.5×comH proximally of the sinutubular junction as shown with the white dashed line in Figure 4. The keys are the same as those in Figure 4.

**Figure 6 bioengineering-10-01219-f006:**
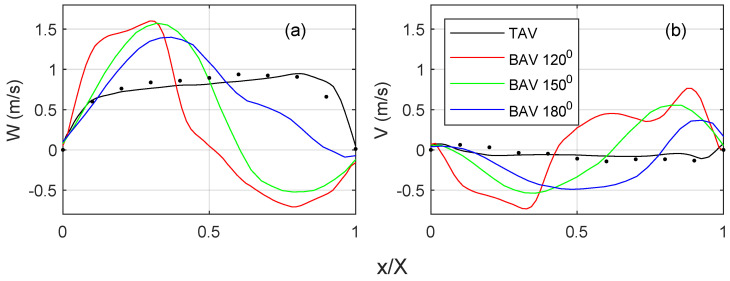
(**a**) Streamwise velocity profile (W) and (**b**) cross-flow velocity profile (V) distribution along the white dashed line that is located proximal to the sinutubular junction, see annotated white dashed line in Figure 5. The *x*-axis of the profile is normalized with the total length, where x/X=0 is towards the convex side and x/X=1 is towards the concave side of the ascending aorta. All cases are for peak systole. The 4D Flow MRI data for the normal TAV cases are shown with black dots.

**Figure 7 bioengineering-10-01219-f007:**
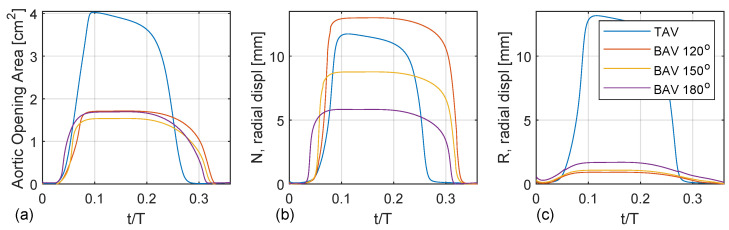
(**a**) Aortic opening area as a function of the cardiac cycle for the TAV and BAV cases. (**b**) Radial displacement of the non-coronary leaflet at the mid-point location of the free edge. (**c**) Radial displacement of the right leaflet at the mid-point location of the free edge.

**Figure 8 bioengineering-10-01219-f008:**
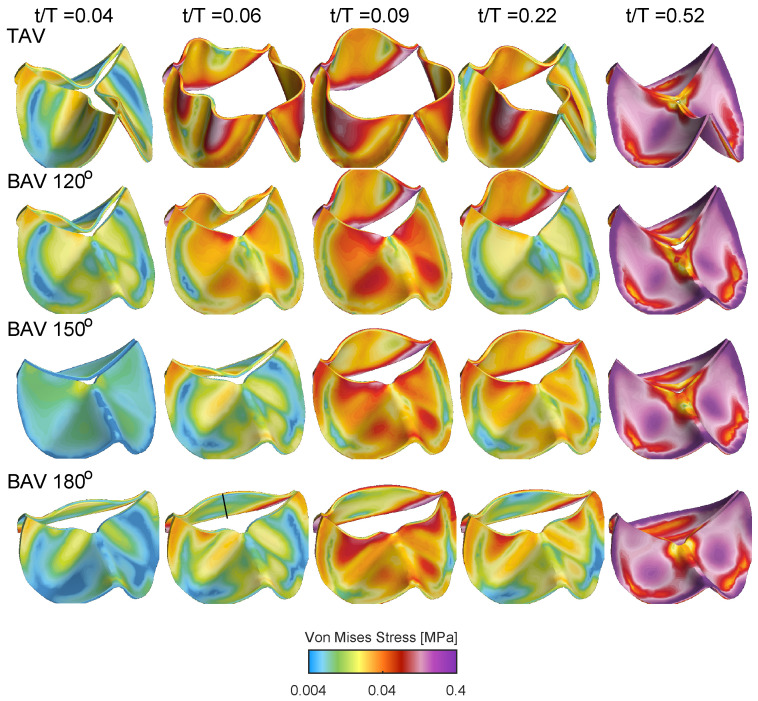
Isometric views showing the von Mises stress distribution during opening and closing for the TAV case and the BAV cases during the cardiac cycle.

**Figure 9 bioengineering-10-01219-f009:**
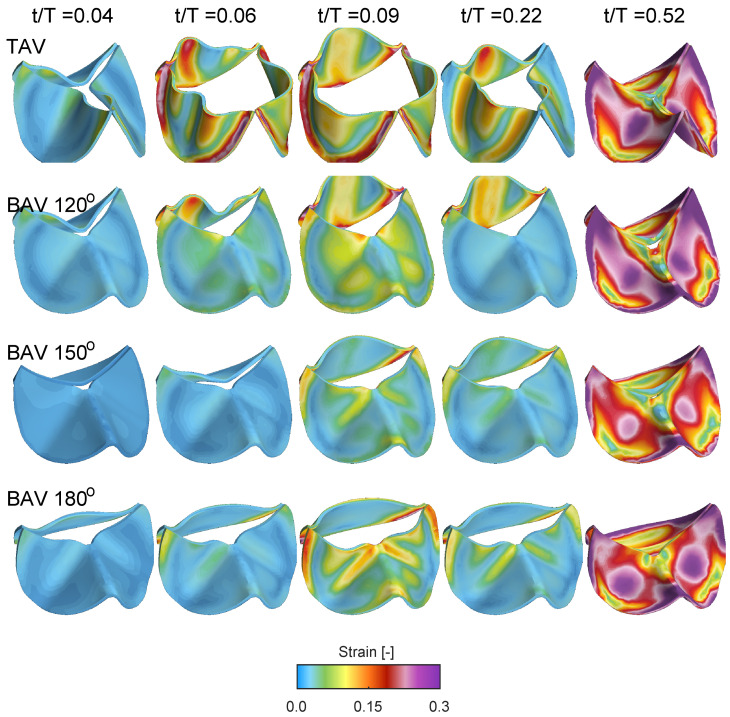
Isometric views showing the Frobenius norm of the strain tensor during opening and closing for the TAV case and the BAV cases with full coronary leaflet fusion during the cardiac cycle. Same keys as in Figure 8.

**Figure 10 bioengineering-10-01219-f010:**
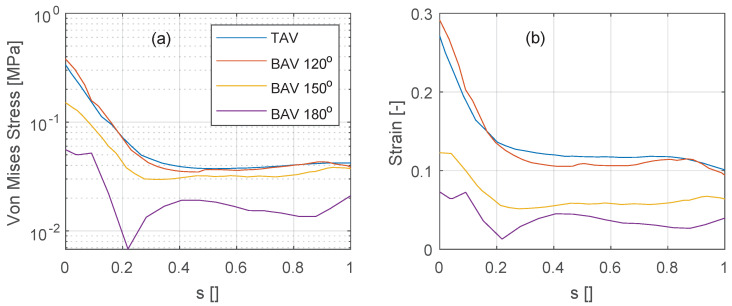
Variation of (**a**) von Mises stress and (**b**) Frobenius norm of the strain tensor along the mid of the non-coronary leaflet between the intersection with the aortic root (s=0) up to the free edge (s=1). Data are presented for the TAV case and the BAV cases.

## Data Availability

Data available on request due to restrictions in repository access.

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
