# Peer review of "Impact of Variation in Commissural Angle between Fused Leaflets in the Functionally Bicuspid Aortic Valve on Hemodynamics and Tissue Biomechanics"

_bioengineering, 2023, doi:10.3390/bioengineering10101219_

Round 1

Reviewer 1 Report

This study aims to know the association between commissural angle of bileaflet aortic valves and aortic blood flow/valvular strain/stress. The author made simulations using CMR with different commissural angles and showed that the commissural angle in bileaflet aortic valve can affect aortic blood flow and valvular strain or stress. This study starts with multiple assumptions and that can lead to many questions.   1. Figure 3 only includes BAV case at angle of 120 deg. It would be better to include better 3-dimentional visualization with multiple angle cases.   2. Regarding the commissural angle and aortic flow, please see this article and compare with that. https://www.nature.com/articles/s41598-021-81845-w   3. The leaflet thickness and mechanical properties (e.g., calcification) is critical in learning the mechanical effects on the valve, especially in congenital bicuspid valve disease as the authors mentioned in the limitation. Multimodal imaging approach would be essential to know better about the outcomes of morphology of bileaflet aortic valve.   4. It would be great if the velocity data can be validated with 4D flow MRI. 

Author Response

Manuscript ID: bioengineering-2638769 rebuttal letter 

The authors are grateful for the comments received from the reviewer. All comments and suggestions have been considered and addressed as part of the revised version of the manuscript. Changes to the manuscript have been highlighted with a red font. 

Sincerely, 

Elias Sundström and Justin T. Tretter 
Stockholm, Sweden, 2023-10-12

Reviewer #1 
This study aims to know the association between commissural angle of bileaflet aortic valves and aortic blood flow/valvular strain/stress. The author made simulations using CMR with different commissural angles and showed that the commissural angle in bileaflet aortic valve can affect aortic blood flow and valvular strain or stress. This study starts with multiple assumptions and that can lead to many questions.

1. Figure 3 only includes BAV case at angle of 120 deg. It would be better to include better 3-dimentional visualization with multiple angle cases.
A: Thank you for this comment. The previous figure is now split into two figures that show the TAV case and all considered BAV cases (i.e. 120, 150, 180 deg). Figure 4 shows the velocity vectors on the vertical plane, and Figure 5 now presents the velocity and streamline distribution on the short-axis cut plane. 

2. Regarding the commissural angle and aortic flow, please see this article and compare with that. https://www.nature.com/articles/s41598-021-81845-w   
A: We appreciate this comment. This article is now cited in the Introduction and we making comparisons with it in the Discussion.  

3. The leaflet thickness and mechanical properties (e.g., calcification) is critical in learning the mechanical effects on the valve, especially in congenital bicuspid valve disease as the authors mentioned in the limitation. Multimodal imaging approach would be essential to know better about the outcomes of morphology of bileaflet aortic valve.   
A: This is a much appreciated remark. This is now added to the limitation.

4. It would be great if the velocity data can be validated with 4D flow MRI. 
A: Figure 6 has been added that quantifies the streamwise and cross-flow velocity profiles at a location proximally of the sinutubular junction for all considered cases at peak systole. The result from the 4D flow MRI protocol for the TAV case has been added and shows a fair agreement. 

Reviewer 2 Report

The authors present a manuscript of great interest on the implementation of new models in valve problems. This manuscript adequately and precisely achieves an intervention with a very novel model. The authors make an adequate approach to the current state of the art, with impact scientific references, which ensures that the proposal is solid and justified. The authors adequately and precisely describe the methodology to be developed, with references that make the study reproducible, complying with the standards and justifying the methodology.

The results are presented clearly and with self-explanatory figures, which allows the reader to integrate the information of the results presented in an adequate and extraordinarily satisfactory way at all times. The results solidly and precisely support the authors' conclusions.

One of the weak points of this manuscript is the discussion; the authors must integrate point 5 into it, with a direct translational vision in clinical care. In my opinion, a comparative table with other studies can make the scope of the mansuscript greater.

The authors should discuss the sample size more precisely, with a precise technical explanation.

The authors should include a graphic summary that allows readers to fully integrate all the interesting dimensions that the authors have raised and that they have satisfactorily achieved.

I congratulate the authors for this great and appropriate manuscript.

Author Response

Manuscript ID: bioengineering-2638769 rebuttal letter 

The authors are grateful for the comments received from the reviewer. All comments and suggestions have been considered and addressed as part of the revised version of the manuscript. Changes to the manuscript have been highlighted with a red font. 

Sincerely, 

Elias Sundström and Justin T. Tretter 
Stockholm, Sweden, 2023-10-12

Reviewer #2
The authors present a manuscript of great interest on the implementation of new models in valve problems. This manuscript adequately and precisely achieves an intervention with a very novel model. The authors make an adequate approach to the current state of the art, with impact scientific references, which ensures that the proposal is solid and justified. The authors adequately and precisely describe the methodology to be developed, with references that make the study reproducible, complying with the standards and justifying the methodology.

The results are presented clearly and with self-explanatory figures, which allows the reader to integrate the information of the results presented in an adequate and extraordinarily satisfactory way at all times. The results solidly and precisely support the authors' conclusions.

1. One of the weak points of this manuscript is the discussion; the authors must integrate point 5 into it, with a direct translational vision in clinical care. In my opinion, a comparative table with other studies can make the scope of the mansuscript greater.
A: Thank you for this comment. The Discussion is now updated with a vision of the clinical significance. 

2. The authors should discuss the sample size more precisely, with a precise technical explanation.
A: In the Limitations we make clear that this is a small study using data from a normal TAV subject validated with 4D Flow MRI. We also make clear that we need to consider a larger cohort with both stenotic and regurgitant valves undergoing surgery in follow-up studies. However, the small sample size offers a systematic assessment of the commissural angle. Due to the agreement with other studies that used 4D Flow MRI, see Hattori et al. (2021), we feel that the conclusions would not change significantly with a larger cohort.

3. The authors should include a graphic summary that allows readers to fully integrate all the interesting dimensions that the authors have raised and that they have satisfactorily achieved.
A: A graphical abstract has been added that visually summarizes the main findings.

I congratulate the authors for this great and appropriate manuscript.
A: This is a much appreciated remark. 

Round 2

Reviewer 1 Report

Previous comments were well addressed in this version.